# Assay validation and determination of in vitro binding of mefloquine to plasma proteins from clinically normal and FIP-affected cats

**Aaron M. Izes**[☯], **Benjamin Kimble**[☯], **Jacqueline M. Norris**[☯], **Merran Govendir**[ID]*[☯]

Sydney School of Veterinary Science, Faculty of Science, The University of Sydney, Sydney, Australia

☯ These authors contributed equally to this work.
* merran.govendir@sydney.edu.au

**Data Availability Statement:** All data are in manuscript.

**Funding:** M.G. and J.M.N. received the following awards from the following funders to undertake

## Abstract

The antimalarial agent mefloquine is currently being investigated for its potential to inhibit feline coronavirus and feline calicivirus infections. A simple, high pressure liquid chromatography assay was developed to detect mefloquine plasma concentrations in feline plasma. The assay's lower limit of quantification was 250 ng/mL. The mean ± standard deviation intra- and inter-day precision expressed as coefficients of variation were 6.83 ± 1.75 and 5.33 ± 1.37%, respectively, whereas intra- and inter-day accuracy expressed as a percentage of the bias were 11.40 ± 3.73 and 10.59 ± 3.88%, respectively. Accordingly, this validated assay should prove valuable for future in vivo clinical trials of mefloquine as an antiviral agent against feline coronavirus and feline calicivirus. However, the proportion of mefloquine binding to feline plasma proteins has not been reported. The proportion of drug bound to plasma protein binding is an important concept when developing drug dosing regimens. As cats with feline infectious peritonitis (FIP) demonstrate altered concentrations of plasma proteins, the proportion of mefloquine binding to plasma proteins in both clinically normal cats and FIP-affected cats was also investigated. An in vitro method using rapid equilibrium dialysis demonstrated that mefloquine was highly plasma protein bound in both populations (on average > 99%).

## Introduction

Successful treatment of feline infectious peritonitis (FIP), a fatal, virulent coronavirus infection affecting predominantly younger cats, remains difficult [1]. Reviews describing the virus responsible for FIP as well as issues with diagnostics and therapeutics are available [1, 2], providing an explanation as to why re-purposing drugs used in human medicine has been largely unsuccessful in treating FIP infected cats [1]. Recently, the use of antiviral therapies to alter the replication of virulent forms of feline coronavirus known as feline infectious peritonitis virus (FIPV) has shown enormous hope for clinicians [3, 4]. However, veterinary access to these antiviral treatments is currently limited, leading to the global emergence of expensive, unregistered versions of these drugs without the necessary quality control assurances [4, 5]. Accordingly, the need for inexpensive, safe, antiviral medications to treat FIPV-infected cats remains.

this research: Winn Feline Foundation 16-023 Australian Companion Animal Health Foundation 005/2016 Feline Health Research Foundation - The University of Sydney IRMA number 183456.

**Competing interests:** No authors have competing interests.

Mefloquine is currently used for both prevention (as a monotherapy) and treatment (either alone, or in combination with artesunate) of chloroquine-resistant *Plasmodium falciparum* malaria in humans [6–8]. It also has been shown to reduce the viral load of FIPV and feline calicivirus (FCV) at low concentrations in infected Crandell Rees feline kidney cells without cytotoxic effects [9, 10]. Using an in vitro assay, our team demonstrated that while mefloquine undergoes some phase I metabolism in cats, it does not undertake phase II glucuronidative metabolism in this species [11]. In anticipation of conducting in vivo clinical trials of mefloquine to treat FIPV-infected cats, mefloquine's plasma concentrations must be monitored to optimise the dosage regimen. Knowledge of the amount of drug bound to plasma proteins is also important when optimising dosage regimens as only the unbound (or free) fraction is therapeutically active [12, 13]. Although the plasma protein binding of mefloquine has been determined to be > 98% in humans [14], its binding percentage in cats has not been reported. This information is particularly germane in any investigation of a potential FIP antiviral as FIP-affected cats have altered concentrations of the plasma proteins albumin (decreased) and globulins (elevated) compared to clinically normal cats [1, 15]. Consequently, the aim of this study was two-fold: first, to develop and validate a high pressure liquid chromatography (HPLC) method to detect mefloquine in feline plasma, and second, to determine the in vitro plasma protein binding of mefloquine in both clinically normal and FIP-affected cats.

## Materials and methods

### Chemicals

Mefloquine hydrochloride and verapamil hydrochloride (as the internal standard [IS]) were purchased from Sigma-Aldrich (Castle Hill, NSW, Australia). HPLC grade methanol, HPLC grade acetonitrile, phosphate buffered saline pH 7.4 (PBS) and triethylamine were purchased from Thermo Fisher Scientific (Macquarie Park, NSW, Australia).

### Chromatographic conditions

Mefloquine and verapamil were quantified in feline plasma based on modified HPLC method [16]. The HPLC system consisted of a Shimadzu LC-20AT delivery unit, DGU-20A3 HT degassing solvent delivery unit, SIL-20A auto injector, SPD-20A UV detector and CTO-20A column oven. Shimadzu LC Solution software version 1.24 (Kyoto, Japan) was used for chromatographic control, data collection and data processing. Chromatographic separation was performed with a Microsorb-MV C18 column (250 mm × 4.6 mm i.d., 5.0 μm; Varian, Mulgrave, VIC, Australia) with a 1.0 mm Optic-guard C18 pre-column (Choice Analytical, Thornleigh, NSW, Australia) under a pressure of 1,900 psi at 25˚C. The isocratic mobile phase consisted of a mixture of 25 mM sodium phosphate buffer with 1.0% triethylamine adjusted to pH 2.5 and acetonitrile and methanol (50:25:25, v/v/v) at a flow rate of 1.0 mL/min. The injection volume was 10 μL per sample. The diode array detector was set at a wavelength of 220 nm. The retention times of verapamil and mefloquine were approximately 8.5 and 14.5 minutes, respectively.

### Sample preparation

Stock solutions of mefloquine and verapamil were prepared in 100% methanol and 100% acetonitrile, respectively, both at a concentration of 1.0 mg/mL. Further working solutions of mefloquine were prepared in 100% methanol to achieve final concentrations of 500, 1,000, 2,500, 5,000, 10,000, 25,000 and 50,000 ng/mL. Similarly, working solutions of verapamil were prepared in 100% acetonitrile to yield final concentrations of 12.5 μg/mL for the validation study

and 5.0 μg/mL for the plasma protein binding study. Stock solutions and working solutions were stored at 4.0°C prior to use. Blank, clinically normal, feline plasma was pooled ($n > 5$) and stored at -20°C prior to use for preparation of standards and quality control (QC) samples. The origin of the plasma samples is described below. Mefloquine standard samples (50, 100, 250, 500, 1,000, 2,500 and 5,000 ng/mL) and three quality control samples (QC) (250, 1,000 and 5,000 ng/mL) were prepared by spiking 10 μL of prepared working solutions of mefloquine into 90 μL of blank feline plasma. The proteins within the plasma samples were extracted through simple protein precipitation. Specifically, 100 μL of acetonitrile, containing either 12.5 μg/mL of the IS for the validation study or 5.0 μg/mL of the IS for the plasma protein binding study, was added to the 100 μL feline plasma samples. The samples were then vortexed and centrifuged at 14,000 × g for ten minutes to remove any particulates. The supernatant was injected into the HPLC system.

## HPLC method validation

Assay selectivity was established by analysing pooled ($n > 5$) blank, clinically normal, feline plasma to ensure that there were no endogenous interference peaks around the retention times of mefloquine and verapamil. Mefloquine concentrations were measured *via* standard curves performed on three replicates of each concentration of mefloquine (100, 250, 500, 1,000, 2,500 and 5,000 ng/mL) on three consecutive days. A weighting factor (1/x) was used to assign the relative importance of the observations in the regression; in this case, to ensure that larger observations were not over-fitted. The theoretical lower limit of detection (LLOD, the lowest analyte concentration that can be distinguished from the assay background noise) and the lower limit of quantification (LLOQ, the lowest concentration at which an analyte can be reliably detected at a specified level of accuracy and precision) were estimated as follows [17]: LLOD = 3.3 x σ /S, where σ is the standard deviation of the y-intercepts from the regression lines and S is the mean slope of the calibration curves. Thereafter, the LLOQ was calculated using the formula LLOQ = 3 x LLOD. Acceptance criteria for the LLOQ was defined as precision with a CV ≤ 15% and accuracy within ± 20% of nominal concentration with repeated analyses [18]. Intra- and inter-day precision, expressed as CV (%), were analysed from triplicates of QC samples (250, 1,000 and 5,000 ng/mL), both within a single day and on three consecutive days, respectively. Intra- and inter-day accuracy, expressed as bias, were determined by a percentage difference between the estimated value and the nominal value of mefloquine as follows: Bias (%) = 100 –(estimated value / nominal value × 100). Absolute recovery of mefloquine and IS were determined by comparing the peak area of pre-spiked plasma samples ($n = 9$) at concentrations of 250, 1,000 and 5,000 ng/mL with corresponding concentrations of mefloquine and IS in mobile phase. Each assay was conducted in triplicate.

The linearity of the data was assessed through linear regression analysis with GraphPad Prism software version 8.1.1 (GraphPad Software, Inc., CA, USA).

## Blood collection

With owner verbal or written consent, blood was collected from both clinically normal and FIP-affected cats. The use of the residual plasma from this patient for this study was approved by The University of Sydney Animal Ethics Committee (protocol: 2016/1027). The animals were considered clinically normal based on inclusion criteria modified from that of Norris *et al*. (2007) [19]. Specifically, the cats designated for inclusion as clinically normal subjects were systemically well and had their blood collected for purposes other than investigating a current illness. Examples of clinically normal cats included those presenting for annual examinations, routine vaccinations or for routine screening tests prior to sedation or general

anaesthesia for de-sexing, grooming, routine dental scaling or radiography post-acute trauma [19]. In contrast, a cat's FIP status was confirmed by direct immunofluorescence of effusion samples and immunohistochemistry on tissue samples (n = 5), or simply immunohistochemistry (n = 5) using methods outlined by Worthing *et al.* (2012) [20]. Additionally, other haematological tests were performed including quantification of albumin: globulin which was consistent with the hyperglobulinaemia observed with cats infected with FIPV [15]. All plasma samples were stored at -20˚C prior to analysis. Plasma samples from both clinically normal and FIP-affected cats were provided by the Paddington Cat Hospital (Sydney, NSW) and the Veterinary Pathology and Diagnostic Service, Sydney School of Veterinary Science.

### Plasma protein binding assay

The in vitro plasma protein binding (PPB) of mefloquine was determined by the rapid equilibrium dialysis (RED) method. The RED assay was performed according to manufacturer's (Thermo Fisher Scientific, Macquarie Park) instructions. Pooled plasma samples ($n > 5$) were assigned to one of two experimental groups, depending on health status, i.e., either clinically normal cat plasma or FIP-affected plasma. Prior to experimentation, the pH of the pooled plasma (approximately pH 8) was adjusted to pH 7.3 [21], to mimic the physiologic pH of cats. Likewise, the total protein (TP), albumin and globulin concentrations of the pooled plasma samples were quantified prior to the assay by the VPDS using a Thermo Fisher Scientific Konelab PRIME 30i analyzer (Scoresby, VIC, Australia) employing conventional protocols.

For each experimental group, two final concentrations of mefloquine (either 5,000 or 10,000 ng/mL) were tested using three replicates for each concentration. These concentrations were selected as they are comparable to the 10 μM concentration (10 μM = 3.78 μg/mL or 3,780 ng/mL) tested by McDonagh *et al.* (2014) [10] in FIPV-infected Crandell Rees feline kidney cells. Accordingly, 300 μL of plasma pre-spiked with mefloquine was added to the red-ringed chamber of the RED device whilst 550 μL of PBS was added to the white-ringed chamber. The unit was covered with sealing tape (Thermo Fisher Scientific, Scoresby, VIC, Australia, Product No. 15036) and incubated on an orbital shaker at 250 rpm for four hours. The temperature was set at 38˚C to mimic the normal body temperature of cats. Following incubation, 100 μL of post-dialysis samples were removed from the red-ringed chambers. These samples were then placed in separate microcentrifuge tubes and precipitated using 100 μL of chilled acetonitrile pre-spiked with the IS (5.0 μg/mL verapamil). After vortexing, the mixtures were centrifuged at $14,000 \times g$ for ten minutes. The supernatant was injected into the HPLC system. Similarly, 200 μL volumes of post-dialysis samples were removed from the white-ringed chambers. These samples also underwent HPLC analysis. All samples were prepared and analysed in triplicate. The PPB of mefloquine was determined by the following equation:

$$\%Free = (Concentration\ buffer\ chamber/Concentration\ plasma\ chamber) \times 100\%$$

$$\%Bound = 100 - \%Free$$

### Statistical analyses

Statistical data analysis was performed using GraphPad Prism software version 8.1.1 (Graph Pad Software, Inc., CA, USA). Prior to parametric testing, a Shapiro-Wilk normality test performed on the mefloquine PPB data indicated a normal distribution. Likewise, an inspection of a normal quantile-quantile (QQ) plot of this data further corroborated its normality. A two-way ANOVA evaluating the effects of health status and drug concentration on mefloquine PPB was performed. Results were considered statistically significant at $p < 0.05$.

## Results

### Method development and optimisation

Based on the UV spectra, the greatest area under the mefloquine peak was at a wavelength of 220 nm. The retention times of the IS and mefloquine were approximately 8.5 and 14.5 minutes, respectively. No endogenous interference was observed at the retention times of mefloquine and the IS. Chromatograms of feline plasma pre-spiked with mefloquine and the IS as well as blank feline plasma are shown in Fig 1.

### HPLC assay validation

**Selectivity.** Pooled blank feline plasma and feline plasma pre-spiked with mefloquine (250, 1,000 and 5,000 ng/mL) and the IS (12.5 µg/mL) were used to check the selectivity of this method. Due to the limited volume of FIP-affected feline plasma available, only blank plasma of clinically normal cats was used. As demonstrated in Fig 1, no endogenous plasma components interfered with elution of analytes.

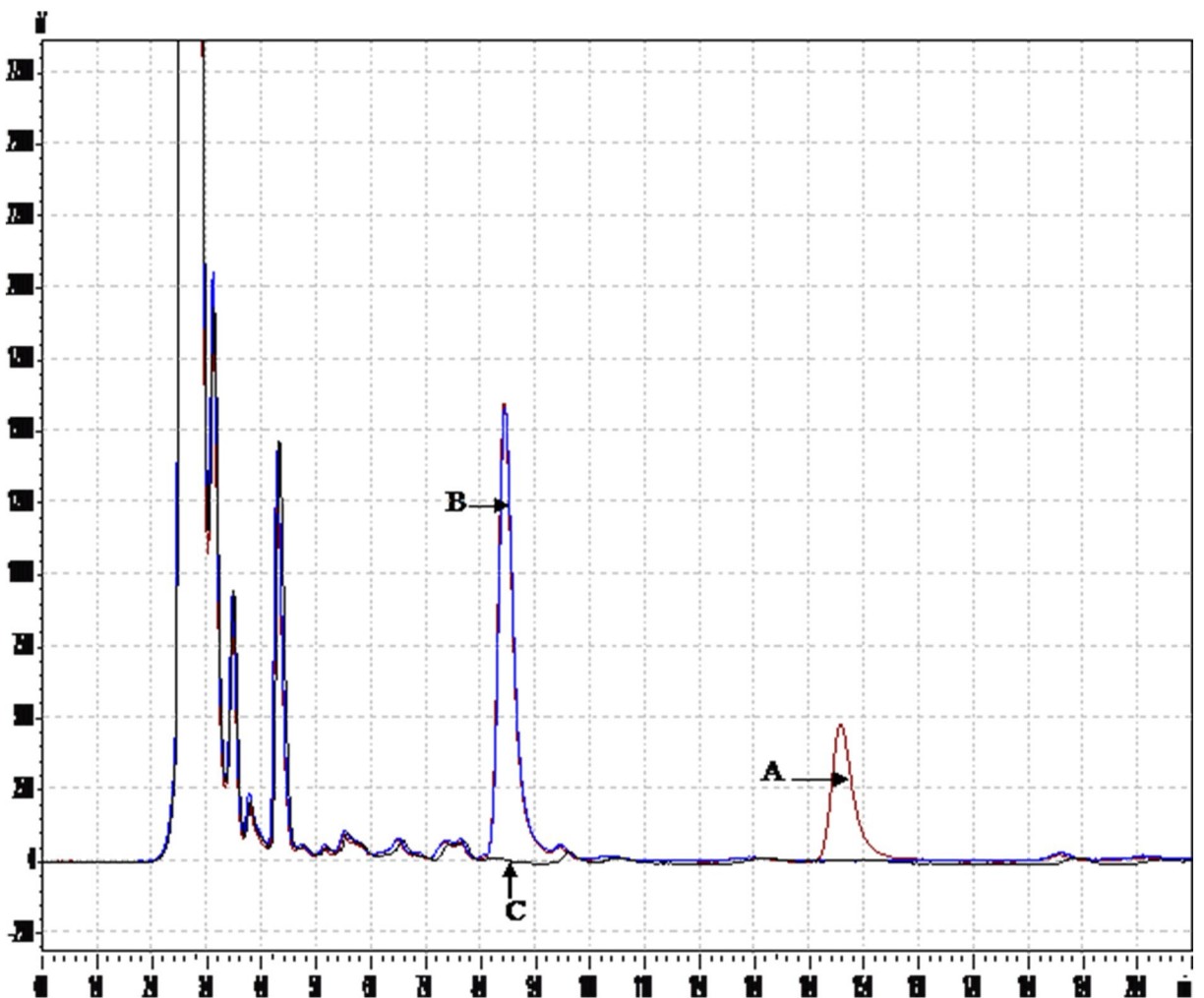

**Fig 1.** "Stacked" chromatograms of 5,000 ng/mL of mefloquine (A, brown trace) and 12.5 µg/mL verapamil as the IS (B, blue trace) and pooled (*n* > 5), blank, clinically normal feline plasma (C, black trace) at the UV wavelength of 220 nm.

**Linearity, LLOD, LLOQ, accuracy and precision.**   The plasma peak ratio (area of mefloquine divided by the IS area) versus the concentration was plotted and determined to be linear for the concentration range used (100 to 5,000 ng/mL). The mean regression standard curves ($n$ = 3) for mefloquine were described as y = 9.374 e-005 x + 0.001417, with a weighting factor of 1/x. The correlation coefficient value ($r^2$) for each curve $\geq$ 0.97. Estimated from standard curves, the LLOD for mefloquine was 20.5 ng/mL and the LLOQ was 61.5 ng/mL. However, as mefloquine concentrations < 250 ng/mL were not accurate (>20% accuracy), the LLOQ was set as 250 ng/mL.

Intra- and inter-day precision expressed as CVs ranged from 5.30 to 8.74% and 3.77 to 6.32%, respectively. Intra- and inter-day accuracy expressed as a percentage of the bias ranged from -12.67 to 13.96% and -15.01 to 9.04%, respectively. These values satisfied the guidelines regarding assay reliability [18]. Intra- and inter-day precision and accuracy are summarised in Table 1.

**Drug recovery from plasma.**   The absolute recovery rates of mefloquine expressed as percentages ± standard deviation (S.D.) from the 250, 1,000, and 5,000 ng/mL QC samples ($n$ = 3) were 107.82 ± 5.93; 99.91 ± 5.58; and 105.21 ± 6.32, respectively. The absolute recovery rate of the IS expressed as a percentage ± S.D. was 123.74 ± 3.77 ($n$ = 9).

**In vitro plasma protein binding of mefloquine in healthy and FIP-affected cats.**   The measured concentrations of total protein, albumin and globulin in the pooled plasma for the respective experimental groups are provided in Table 2. Although reference intervals (RI) vary depending on the laboratory and the methodology used for quantification, reference intervals from Cornell University's Animal Health Diagnostic Center [22] were used as none were provided by the testing laboratory.

The results of in vitro plasma protein binding of mefloquine in clinically normal and FIP-affected cats are provided in Table 3.

A significant difference was found between the plasma protein binding of mefloquine in clinically normal and FIP-affected cats ($p$ = 0.0004), even though mefloquine was determined to be highly plasma protein bound (on average > 99%) in both groups. Moreover, a significant interaction effect between health status and mefloquine concentration was identified ($p$ = 0.0007). In contrast, no significant difference was discovered between the plasma protein binding of the two concentrations of mefloquine ($p$ = 0.11).

## Discussion

There has been recent interest in the suitability of mefloquine as an antiviral treatment for FIP [9, 10, 23]. The majority of publications that detect and quantify mefloquine use liquid chromatography. Liquid chromatography demonstrates good sensitivity (i.e., it can detect low concentrations of mefloquine), is highly specific and the machinery is affordable for veterinary diagnostic laboratories. Although an HPLC assay to detect mefloquine in an in vitro matrix has been reported [23], in contrast, this study describes an assay to quantify mefloquine

**Table 1. Intra- and inter-day precision and accuracy for the QC samples (250, 1000 and 5000 ng/mL of mefloquine) tested in triplicate for each concentration.**

| Nominal concentration (ng/mL) | Measured concentrations (ng/mL) | | Precision (CV, %) | | Accuracy (Bias, %) | |
|---|---|---|---|---|---|---|
| | Intra-day | Inter-day | Intra-day | Inter-day | Intra-day | Inter-day |
| 250 | 215.11 ± 18.79 | 230.67 ± 14.41 | 8.74 | 6.32 | 13.96 | 7.73 |
| 1,000 | 1,126.73 ± 59.74 | 1,150.09 ± 42.83 | 5.30 | 3.77 | -12.67 | -15.01 |
| 5,000 | 4,621.15 ± 298.44 | 4,547.80 ± 266.95 | 6.45 | 5.89 | 7.58 | 9.04 |
| Mean (absolute numbers) ± standard deviation | | | 6.83 ± 1.75 | 5.33 ± 1.37 | 11.40 ± 3.73 | 10.59 ± 3.88 |

**Table 2. Total protein, albumin (A), globulin (G) and A: G ratio in pooled plasma ($n > 5$) for both clinically normal and FIP-affected cats.**

| Health status | Total Protein (RI 66–84 g/L) | Albumin (RI 32–43 g/L) | Globulins (RI 29–47 g/L) | A:G ratio (RI 0.8–1.5) |
|---|---|---|---|---|
| Clinically normal | 72.19 | 32.17 | 40.02 | 0.80 |
| FIP-affected | 86.05 | 18.73 | 67.32 | 0.28 |

concentrations in feline plasma. The assay described here has a straightforward drug extraction procedure and only requires a UV detector. Based on McDonagh *et al.* (2014) [10], 10 μM (3.78 μg/mL) mefloquine has demonstrated in vitro antiviral activity against FIPV. As this HPLC assay can reliably detect < 1 μM (378 ng/mL), it should prove valuable in future in vivo clinical trials of mefloquine administered to FIPV-affected cats.

To the authors' knowledge, this is the first report of mefloquine PPB in a non-human species. This study provides the novel finding that mefloquine is highly plasma protein bound in cats. In humans, mefloquine is also highly plasma protein bound (> 98%) [14], which may account, in part, for its long half-life of approximately three weeks in healthy human subjects [24]. High plasma protein binding can result in the drug-protein complex acting as a reservoir for the physiologically active free drug concentration and consequently prolonging its duration of action [12, 25]. Likewise, it should not be assumed that a drugs' PPB is constant across species [25]. In a comparative species study investigating the plasma protein binding of 574 compounds, drugs tend to be slightly more bound in human plasma proteins in comparison to the plasma proteins of rats, mice and dogs [26]. Some of the compounds that showed significant interspecies differences in plasma protein binding included diazepam (98% PPB in humans versus 84.8% in rats), prazosin (94.7% PPB in humans versus 65% in rats versus 63% in dogs) and sildenafil (93.6% PPB in humans versus 74% in dogs) [26].

Diseases, such as FIP, can alter the concentrations of plasma proteins and in turn impact upon a drug's plasma protein binding and ultimately its therapeutic efficacy [27]. For example, in humans, the proteins albumin and $\alpha_1$-acid glycoprotein (AAG) provide the largest contribution to the protein binding of drugs [27, 28]. Yet, with acute falciparum malaria, serum concentrations of AAG in non-immune human patients increase two-fold within 24 hours whereas plasma levels of albumin decrease by 30% [29]. Likewise, in FIP-affected cats, common serum protein abnormalities may include elevated AAG [30] and globulin levels [1, 15] as well as decreased albumin levels and a low A: G ratio [15]. Here, as described in the literature, the confirmed FIP plasma samples demonstrated elevated globulin and decreased albumin levels as well as a low A: G ratio.

Mefloquine has a high affinity for AAG binding, preferentially binding to AAG over albumin [27]. Thus, if more AAG is present in FIP-affected cats, it may be preferentially bound by mefloquine. Yet, small changes in the unbound drug fraction of highly protein bound drugs can have a significant therapeutic impact [28]. For example, the reduction of PPB from 99.9% to 99.8% can lead to a doubling of therapeutically active unbound drug concentration in plasma [28]. Here, although a significant difference was found between the plasma protein binding of mefloquine in clinically normal and FIP-affected cats, due to the unknown

**Table 3. In vitro plasma protein binding (mean ± S.D.) of two concentrations (5,000 and 10,000 ng/mL) of mefloquine as determined by rapid equilibrium dialysis in clinically normal and FIP-affected cats.** Three replicates were performed for each group.

| Concentration (ng/mL) | Clinically normal (PPB %) | FIP-affected (PPB %) |
|---|---|---|
| 5,000 | 99.51 ± 0.13 | 99.55 ± 0.14 |
| 10,000 | 98.95 ± 0.18 | 99.84 ± 0.06 |

biological variability of the assay, it is likely that this difference is equivocal. Fundamentally, it is difficult to generalise from two pooled plasma samples to two populations of individuals since the biological variability in each population is unknown. The variability (standard deviation) used to evaluate differences in PPB were derived from the assay method and do not represent biological differences. Yet, it is also unlikely that further in vitro mefloquine feline plasma protein binding studies will resolve this issue. On the contrary, observing the clinical response of mefloquine in FIP-affected animals ultimately may decide mefloquine's therapeutic efficacy in this patient population.

In vitro studies to quantify plasma protein binding have some limitations. Whilst the major advantage of the RED method include its speed, simplicity, reliability and cost-effectiveness [31], its drawbacks, which also serve as limitations to this study, involve nonspecific membrane binding of the drug [31] as well as changes in oncotic pressure leading to an overestimation of unbound drug concentration [32]. Moreover, in vitro PPB may not mirror in vivo PPB in the live animal as the structure of the plasma proteins and their affinity to bind substrates can vary with age, disease and /or presence of competing endogenous and exogenous compounds such as dietary constituents or other therapeutic drugs [33, 34]. Furthermore, the drug-protein dissociation rate, which can also greatly affect highly PPB drugs [35], was not determined in this study. Likewise, an additional limitation to this project was the scarcity of confirmed FIP-affected plasma as this impacted upon its availability for use in both the HPLC validation and PPB protocols. Finally, given mefloquine's purported mechanism of action as a schizonticide, another potential limitation concerns the effects of the intraerythrocytic accumulation of mefloquine on plasma protein binding. However, previous studies have demonstrated that red blood cells do not serve as a significant depot for mefloquine [36, 37].

Recently, the broad spectrum coronavirus protease inhibitor, GC376 [38], and the adenosine nucleoside analogue, GS-441524 [3, 4] have been shown to be safe and efficacious antiviral agents for use against FIPV. Yet, as neither of these agents has obtained registration for veterinary use, investigations into more readily available drugs with anti-FIPV activity, such as mefloquine, are urgently required for this invariably fatal disease. This study has validated an accurate and reliable assay to detect mefloquine in feline plasma and demonstrated that mefloquine is highly plasma protein bound in both clinically normal and FIP-affected cats. Further studies describing mefloquine's pharmacokinetic profile in the cat should be progressed.

## Acknowledgments

Professor Michael Court (Washington State University, College of Veterinary Medicine) provided invaluable insight into the interpretation of the mefloquine plasma protein binding results. We also thank the Veterinary Pathology and Diagnostics Services, Sydney School of Veterinary Science and Dr. Randolph Baral (The Paddington Cat Hospital, Sydney, NSW), for providing the feline plasma samples.

## Author Contributions

**Conceptualization:** Jacqueline M. Norris, Merran Govendir.

**Formal analysis:** Aaron M. Izes.

**Funding acquisition:** Jacqueline M. Norris, Merran Govendir.

**Investigation:** Aaron M. Izes, Benjamin Kimble.

**Methodology:** Benjamin Kimble.

**Project administration:** Jacqueline M. Norris, Merran Govendir.

**Resources:** Jacqueline M. Norris, Merran Govendir.

**Supervision:** Benjamin Kimble, Jacqueline M. Norris, Merran Govendir.

**Validation:** Aaron M. Izes, Benjamin Kimble.

**Writing – original draft:** Aaron M. Izes.

**Writing – review & editing:** Benjamin Kimble, Jacqueline M. Norris, Merran Govendir.

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
