## [Decision Letter · Decision Letter 0]

11 May 2020

PONE-D-20-08739

Assay validation and determination of in vitro binding of mefloquine to plasma proteins from clinically normal and FIP-affected cats

PLOS ONE

Dear Dr Govendir

Thank you for submitting your manuscript to PLOS ONE. After careful consideration, we feel that it has merit but does not fully meet PLOS ONE’s publication criteria as it currently stands. Therefore, we invite you to submit a revised version of the manuscript that addresses the points raised during the review process.

Many thanks for submitting your manuscript to PLOS One

Your manuscript was reviewed by three expert reviewers, who have recommended that some modifications be made prior to acceptance.

If you could write a response to reviewers, that will expedite things when the manuscript is re-submitted

I wish you the best of luck with your revisions

Hope you are keeping safe and well at this difficult time

Thanks

Simon

We would appreciate receiving your revised manuscript by Jun 25 2020 11:59PM. To enhance the reproducibility of your results, we recommend that if applicable you deposit your laboratory protocols in protocols.io, where a protocol can be assigned its own identifier (DOI) such that it can be cited independently in the future. For instructions see: http://journals.plos.org/plosone/s/submission-guidelines#loc-laboratory-protocols

We look forward to receiving your revised manuscript.

Kind regards,

Simon Russell Clegg, PhD

Academic Editor

PLOS ONE

2. In your methods section please state whether informed consent of cat owners was obtained, and if so, if consent was written or verbal.

Reviewers' comments:

Reviewer's Responses to Questions

**Comments to the Author**

1. Is the manuscript technically sound, and do the data support the conclusions?

Reviewer #1: Yes

Reviewer #2: Yes

Reviewer #3: Yes

2. Has the statistical analysis been performed appropriately and rigorously? 

Reviewer #1: I Don't Know

Reviewer #2: Yes

Reviewer #3: Yes

3. Have the authors made all data underlying the findings in their manuscript fully available?

Reviewer #1: Yes

Reviewer #2: No

Reviewer #3: Yes

4. Is the manuscript presented in an intelligible fashion and written in standard English?

Reviewer #1: Yes

Reviewer #2: Yes

Reviewer #3: Yes

5. Review Comments to the Author

Reviewer #1: The manuscript entitled “Assay validation and determination of in vitro binding of mefloquine to plasma proteins from clinically normal and FIP-affected cats” by Izes et al., describes the development of quantitative assay for mefloquine hydrochloride by HPLC. Verapamil hydrochloride was used as the internal standard. The validated assay was then applied to quantitate amount of plasma protein bound mefloquine hydrochloride from normal and FIPV infected cats. The plasma samples were from clinically normal and FIP-affected cats. The researchers pooled 5 samples of each clinical type of plasmas and differentiated them by using albumin:globulin (A:G) ratio. Mefloquine was spiked into the plasma samples and plasma protein bound mefloquine hydrochloride was quantitated by rapid equilibrium dialysis (RED) method. A:G ratio is a non-specific and rough method to diagnose FIP cats based on the fact that immunoglobulin was exacerbated in FIP-affected cats. The authors should use RT-PCR to detect the viral nucleic acid in the plasma to confirm the diagnosis. Majority of contents in the manuscript are related to the assay development and validation which is not interesting as HPLC is not new technology. The authors did not compare this assay with other methods and show how attractive it is. In addition, not much assay application was described.

Reviewer #2: PLOS ONE

Manuscript number: PONE-D-20-08739

Reviewer’s comments:

This manuscript is described that assay validation and determination of binding of mefloquine to plasma proteins. The manuscript is well written as a whole. However, I have a minor question on this manuscript. Therefore, please ask the question.

Minor comments;

The authors used mefloquine and verapamil for this assay but I wonder why you used verapamil. You did not show verapamil data in the Figs. and Tables in the manuscript. Therefore, you should explain the research object to use verapamil and show any data on verapamil studies.

Reviewer #3: In this paper, the authors propose a method for measuring mefloquine binding to feline plasmatic proteins, which is fundamental to assess the in vivo efficacy of this drug in cats with feline infectious peritonitis (FIP). Although the study presents some limitations, mainly related to the in-vivo adaptation of in-vitro model, it deserves publication after few issues are addressed.

Introduction. A brief introduction of the virus, its characteristics and the caused disease (FIP) would be beneficial to the general audience of the journal. See Decaro and Lorusso (Vet. Microbiol. 2020, https://doi.org/10.1016/j.vetmic.2020.108693) for an overview.

Blood collection. The authors should provide details on how many FIP cases were diagnoses using immunofluorescence assay (IFA) on effusions and how many using immunohistochemistry, which is the gold standard for FIP diagnosis. The authors should consider that IFA is not the gold standard and that an IFA-based diagnosis should be supported by additional evidence according to the European Advisory Board of Cat Disease recommendations, including Rivalta’s test, total proteins, albumin/globulin ratio, total leukocyte counts and identity of cells (Lorusso et al., Res Vet Sci. 2019 Aug;125:421-424).

Discussion. A more detailed discussion about the comparative analysis with COVID-19 therapy should be presented.

6. PLOS authors have the option to publish the peer review history of their article (what does this mean?). If published, this will include your full peer review and any attached files.

Reviewer #1: No

Reviewer #2: No

---

## [Author Response · Author response to Decision Letter 0]

1 Jun 2020

The responses to the Reviewers has been provided as a attachment

---

## [Decision Letter · Decision Letter 1]

14 Jul 2020

Assay validation and determination of in vitro binding of mefloquine to plasma proteins from clinically normal and FIP-affected cats

PONE-D-20-08739R1

Dear Dr. Govendir

We’re pleased to inform you that your manuscript has been judged scientifically suitable for publication and will be formally accepted for publication once it meets all outstanding technical requirements.

Kind regards,

Simon Clegg, PhD

Academic Editor

PLOS ONE

Additional Editor Comments:

Many thanks for resubmitting your manuscript to PLOS One

The manuscript was sent to the same two reviewers as last time.

Unfortunately one was unavailable, but the other was happy. I therefore provided the 2nd review for the manuscript, and as all comments were addressed and the manuscript reads well, I have recommended it for publication

It was a pleasure working with you, and I wish you all the best for your future research.

Hope you are keeping safe and well in these difficult times.

Thanks

Simon

Reviewers' comments:

Reviewer's Responses to Questions

**Comments to the Author**

1. If the authors have adequately addressed your comments raised in a previous round of review and you feel that this manuscript is now acceptable for publication, you may indicate that here to bypass the “Comments to the Author” section, enter your conflict of interest statement in the “Confidential to Editor” section, and submit your "Accept" recommendation.

Reviewer #2: All comments have been addressed

2. Is the manuscript technically sound, and do the data support the conclusions?

Reviewer #2: Yes

3. Has the statistical analysis been performed appropriately and rigorously? 

Reviewer #2: Yes

4. Have the authors made all data underlying the findings in their manuscript fully available?

Reviewer #2: Yes

5. Is the manuscript presented in an intelligible fashion and written in standard English?

Reviewer #2: Yes

6. Review Comments to the Author

Reviewer #2: Now this manuscript has been improved for my assistance, therefore, I will allow the manuscript to be published for POS ONE.

7. PLOS authors have the option to publish the peer review history of their article (what does this mean?). If published, this will include your full peer review and any attached files.

Reviewer #2: No

---

## [Editor Report · Acceptance letter]

17 Jul 2020

PONE-D-20-08739R1 

Assay validation and determination of in vitro binding of mefloquine to plasma proteins from clinically normal and FIP-affected cats 

Dear Dr. Govendir:

I'm pleased to inform you that your manuscript has been deemed suitable for publication in PLOS ONE. Congratulations! Your manuscript is now with our production department. 

Kind regards, 

on behalf of

Dr. Simon Clegg 

Academic Editor

PLOS ONE